# Diagnostic Accuracy of Slow-Capillary Endobronchial Ultrasound Needle Aspiration in Determining PD-L1 Expression in Non-Small Cell Lung Cancer

Lina Zuccatosta [1,*], Federico Mei [1,2], Michele Sediari [1], Alessandro Di Marco Berardino [1], Martina Bonifazi [1,2], Francesca Barbisan [3], Gaia Goteri [3], Stefano Gasparini [1,2] and Francesca Gonnelli [1]

[1] Pulmonary Diseases Unit, Azienda Ospedali Riuniti, Via Conca 71, 60126 Ancona, Italy
[2] Pulmonary Disease Unit, Dipartimento di Scienze Biomediche e Sanità Pubblica, Polytechnic University of Marche, Via Conca 71, 60126 Ancona, Italy
[3] Pathological Anatomy Institute, Polytechnic University of Marche, Via Conca 71, 60126 Ancona, Italy
[*] Correspondence: linazuccatosta@tiscali.it

**Highlights:**

 **What are the main findings?**

- Endobronchial ultrasound transbronchial needle aspiration (EBUS-TBNA) with slow-pull capillary aspiration has been found to be an accurate tool to obtain adequate samples for PD-L1 evaluation in patients with non-small cells lung cancer (NSCLC) and mediastinal lymphadenopathies.

 **What is the implication of the main finding?**

- EBUS-TBNA with no-suction technique (e.g., slow-pull capillary aspiration) can be used not only to get a definite diagnosis of lung cancer in patients with mediastinal lymphadenopathies but also to assess PD-L1 expression and, consequently, to guide treatment decisions (i.e., whether use immunotherapy) in patients with NSCLC.

**Abstract:** Introduction: The role of EBUS-TBNA in the diagnosis and staging of lung cancer is well established. EBUS-TBNA can be performed using different aspiration techniques. The most common aspiration technique is known as "suction". One alternative to the suction technique is the slow-pull capillary aspiration. To the best of our knowledge, no studies have assessed the diagnostic yield of slow-pull capillary EBUS-TBNA in PD-L1 amplification assessment in NSCLC. Herein, we conducted a single-centre retrospective study to establish the diagnostic yield of slow-pull capillary EBUS-TBNA in terms of PD-L1 in patients with NSCLC and hilar/mediastinal lymphadenopathies subsequent to NSCLC. Materials and Methods: Patients with hilar and/or mediastinal lymph node (LN) NSCLC metastasis, diagnosed by EBUS-TBNA between January 2021 and April 2022 at Pulmonology Unit of "Ospedali Riuniti di Ancona" (Ancona, Italy) were enrolled. We evaluated patient characteristics, including demographic information, CT scan/ FDG-PET features and final histological diagnoses, including PD-L1 assessment. Results: A total of 174 patients underwent EBUS-TBNA for diagnosis of hilar/mediastinal lymphadenopathies between January 2021 and April 2022 in the Interventional Pulmonology Unit of the "Ospedali Riuniti di Ancona". Slow-pull capillary aspiration was adopted in 60 patients (34.5%), and in 30/60 patients (50.0%) NSCLC was diagnosed. EBUS-TBNA with slow-pull capillary aspiration provided adequate sampling for molecular biology and PD-L1 testing in 96.7% of patients (29/30); in 15/29 (51.7%) samples with more than 1000 viable cells/HPF were identified, whereas in 14/29 (48.3%) samples contained 101–1000 viable cells/HPF. Conclusion: These retrospective study shows that slow-pull capillary aspiration carries an excellent diagnostic accuracy, almost equal to that one reported in literature, supporting its use in EBUS-TBNA for PD-L1 testing in NSCLC.

**Keywords:** EBUS-TBNA; PD-L1; NSCLC; lung cancer; lymph nodes

## 1. Introduction

The role of endobronchial ultrasound needle aspiration (EBUS-TBNA) in the diagnosis and staging of lung cancer is well established [1]. EBUS-TBNA has significantly increased the diagnostic yield, reaching 90–95% [2–4].

EBUS-TBNA can be performed using different aspiration techniques. The most common aspiration technique is known as "suction". A negative pressure is produced by applying an ex-vacuum EBUS-TBNA dedicated syringe to the needle [5]. One alternative to the suction technique is the slow-pull capillary aspiration, in which no suction is carried out and a slow retraction of the inner stylet is performed [6,7]. Slow-pull capillary aspiration technique has been thought to be less traumatic for tissues with minor blood contamination of the samples [5–7].

Only a few randomized controlled trials have demonstrated the non-inferiority of no-suction aspiration EBUS-TBNA in terms of diagnostic accuracy, but none of them have investigated the adequacy for molecular biology assessment in non-small cell lung cancer (NSCLC).

Immunotherapy with antibodies that prevents the binding of Programmed Death Ligand 1 (PD-L1) to its receptor (PD-L1R) has become crucial in a subset of patients with advanced NSCLC, as it has dramatically improved disease-free and overall survival, through innate immune system response against the cancer: binding of PD-1 ligand to its receptors can prevent an innate cytotoxic T-cell response against tumor by inhibiting kinases that are involved in T-cell activation [8]. Immunotherapy with anti–PD-L1 or anti–PD-1 antibodies unleashes the innate immune system to react to the tumor growth. Recently, several anti-PD-1 and anti-PD-L1 agents have been approved by the Food and Drug Administration and the European Medical Association for patients with metastatic NSCLC both in the first- and second-line settings.

In this context the degree of PD-L1 expression in the primary tumour represents a strong predictor of response to immunotherapy, thus becoming determinant in the diagnostic pathway of advanced NSCLC.

To the best of our knowledge, no studies have assessed the diagnostic yield of slow-pull capillary EBUS-TBNA in PD-L1 amplification assessment in NSCLC.

Herein, we conducted a single-centre retrospective study to establish the diagnostic yield of slow-pull capillary EBUS-TBNA in terms of PD-L1 in patients with hilar/mediastinal lymphadenopathies subsequent to NSCLC.

## 2. Materials and Methods

### 2.1. Patients

Patients with hilar and/or mediastinal lymph node (LN) NSCLC metastasis, diagnosed by EBUS-TBNA between January 2021 and April 2022 at Pulmonology Unit of "Ospedali Riuniti di Ancona" (Ancona, Italy) were consecutively enrolled.

Inclusion criteria were: age >18; hilar and/or mediastinal LN NSCLC metastasis diagnosed by EBUS-TBNA; signed informed consent provided by the patient.

Exclusion criteria were: contraindication to bronchoscopy in accordance to International Guidelines [9,10]; failure of the EBUS-TBNA procedure; incomplete patient information.

LN stations were classified in accordance with the LN map produced by the International Association for the Study of Lung Cancer [11]. We evaluated patient characteristics, including demographic information, CT scan/fluoro-desossi-glucose-positron-emission-tomography (FDG-PET) features [12–15], and final histological diagnoses, including PD-L1 assessment.

### 2.2. EBUS-TBNA Procedure

In all patients, EBUS-TBNA was performed by experienced bronchoscopists (LZ, SG, FM, MS, ADMB). All patients received topical anesthesia (lidocaine2%), and moderate sedation (midazolam 0.035 mg/kg + fentanyl 0.00035 mg/kg) while their vital signs and arterial oxygen saturation were monitored continuously during the whole operation. A

preliminary bronchoscopic exploration of central airways was carried out in all the patients. EBUS-TBNA was performed using an EBUS-scope (Olympus BF-UC-180F, Olympus Corporation, Tokyo, Japan). Following the detection of an enlarged LN by EBUS, color doppler ultrasound was used to avoid vascular structures.

The TBNA biopsies were performed using a 22-gauge needle (Olympus NA-201SX-4022).

The slow-pull capillary technique was performed as follows: after identification and measurement of the target, a needle was used to puncture the LN mass with the stylet in place. Immediately after puncture, the stylet was pushed onto the target to remove the presence of tracheo-bronchial cells in the tip of the needle. Subsequently, the needle was fanned through the target LN, and 15 to-and-fro movements were performed under continuous ultrasonic monitoring. At the same time, the stylet was slowly and continuously pulled to create weak negative pressure. The material was firstly smeared on clean glass slides: one smear was immediately stained by a rapid method (Hemacolor; Merck & Co, Inc., Readington, NJ, USA) for rapid-on-site-evauation (ROSE), and the other slides were wet fixed in 95% ethanol and processed later for definitive cytologic evaluation; then 3 other needle passes were carried out and the material was flushed in 10% neutral-buffered formalin for cell-block [16].

### 2.3. Pathological Evaluation of Samples

The samples acquired using the slow-pull capillary technique were processed for histopathological evaluation and analysed by an expert Pathologist (FB).

The final diagnosis of NSCLC was based on clinical context, hematoxilin/eosin staining, and other biomarkers. In particular, thyroid transcription factor 1 (TTF1) and p40 were tested to distinguish between squamous cell carcinoma and adenocarcinoma; chromogranin A and synaptophysin were used to identify neuroendocrine differentiation in small cell lung cancer (SCLC)/large cell neuroendocrine carcinoma (LCNEC). PD-L1 was tested to establish whether or not immunotherapy is feasible in NSCLC. PD-L1 expression at the cytoplasmic membrane was evaluated only in cancer cells [17]. PD-L1 testing was not routinely performed in non-neoplastic cells and in non-NSCLC cells as in these clinical scenarios its assessment does not provide any additional information and does not guide the decision-making process [16,17].

PD-L1 immunohistochemistry 22C3 pharmDx Autostainer Link 48 (Agilent Technologies, Inc. Santa Clara, CA, USA) immunohistochemical assay was used to assess PD-L1 expression in formalin-fixed, paraffin embedded tissue [18,19].

The quality was evaluated noting the number of viable cells (100–1000/HPF vs. >1000/HPF). Samples were considered adequate for PD-L1 staining if they contained more than 100 evaluable cancer cells; PD-L1 expression is assessed as tumour proportional score (TPS) and PD-L1 positivity was established as PD-L1 > 1% [20].

### 2.4. Statistical Analysis

Quantitative variables were expressed as mean and standard deviation or median and interquartile range, as appropriate. Qualitative variables were expressed as absolute frequency and/or its corresponding percentage.

Quantitative variables were compared using the Mann-Whitney U test. Qualitative variables were compared using the Chi-Square test.

Diagnostic yield was defined as the sum of patients with adequate samples for molecular biology divided by the total number of patients enrolled in the study.

Sensitivity analyses were performed to evaluate whether the diagnostic accuracy for PD-L1 was influenced by the following confounders: sex, ethnicity, smoking status, performance status, LN station, LN endosonographic features and histologic subtype (squamous vs. non-squamous).

Significance was set at $p < 0.05$. STATA v.14 was used to carry out all the analyses.

### 3. Results

A total of 174 patients underwent EBUS-TBNA for diagnosis of hilar/mediastinal lymphadenopathies between January 2021 and April 2022 in the Interventional Pulmonology Unit of the "Ospedali Riuniti di Ancona". Among them, slow-pull capillary aspiration was adopted in 60 patients (34.5%), and in 30/60 patients (50.0%) NSCLC was diagnosed using only this aspiration technique. These patients were included into the study. Table 1. reports the main demographic, clinical, imaging, and histopathological data of the enrolled patients (*n* = 30).

**Table 1.** Demographic, clinical, and imaging data of the study's participants (*n* = 30).

| Age (Mean, SD) | 69.3, 10.2 |
|---|---|
| Female sex (*n*, %) | 12, 40.0% |
| Ethnicity, white Caucasian (*n*, %) | 30, 100% |
| Smoking habit | |
| -     Current smoker (*n*, %) | 0 |
| -     Former smoker (*n*, %) | 24, 80.0% |
| -     Never smoked (*n*, %) | 6, 20.0% |
| Site of sampled lymph nodes | |
| -     Left hilar (10 L) (*n*, %) | 2, 6.0% |
| -     Left inferior paratracheal (4 L) (*n*, %) | 3, 10.0% |
| -     Left interlobar (11 L) (*n*, %) | 0 |
| -     Right hilar (10 R) (*n*, %) | 1, 3.0% |
| -     Right inferior paratracheal (4 R) (*n*, %) | 11, 36.7% |
| -     Right interlobar (11 R) (*n*, %) | 0 |
| -     Subcarinal (7) (*n*, %) | 13, 43.3% |
| CT size of sampled lymph nodes, cm (mean, SD) | 2.4, 0.9 |
| CT shape of sampled lymph nodes | |
| -     Rounded (*n*, %) | 21, 70.0% |
| -     Ovalar (*n*, %) | 9, 30.0% |
| Endosonographic evidence of necrosis (*n*, %) | 5, 16.7% |
| SUV at FDG-PET scan (median, IQR) | 10.8, 9.2–22.4 |
| Histologic subtype | |
| -     Adenocarcinoma (*n*, %) | 27 (90.0%) |
| -     Squamous cell carcinoma (*n*, %) | 3 (10.0%) |

CT: computed tomography, FDG: fluoro-desossi-glucose, IQR: interquartile-range, PET: positron-emission tomography, SD: standard deviation, SUV: standardized uptake value.

EBUS-TBNA with slow-pull capillary aspiration provided adequate sampling for molecular biology and PD-L1 testing in 96.7% of patients (29/30); in 15/29 (51.7%) samples with more than 1000 viable cells/HPF were identified, whereas in 14/29 (48.3%) samples contained 101–1000 viable cells/HPF.

The likelihood of a successful PD-L1 assay did not vary according to sex, LN station, LN endosonographic features, histologic subtype (squamous vs. non-squamous). Conversely, a significantly higher chance of obtaining samples adequate for PD-L1 assessment was found in former/current smokers than in non-smokers (Table 2).

**Table 2.** Diagnostic accuracy of EBUS-TBNA with slow-pull capillary aspiration for molecular biology and PD-L1 testing in stratified analyses (*n* = 30).

| | Diagnostic Yield of Obtaining Samples Adequate for Molecular Biology | *p* Value |
|---|---|---|
| Sex | | |
| Male | 18/18, 100% | 0.21 |
| Female | 11/12, 91.7% | |
| Smoking status | | |
| Never smoked | 5/6, 83.3% | 0.04 |
| Past/current smoker | 24/24, 100.0% | |
| LN station | | |
| 7 | 12/13, 92.3% | 0.51 |
| 4 R | 11/11, 100% | |
| 2 R, 2 L, 4 L, 10 R, 10 L, 11 R, 11 L | 6/6, 100% | |
| Endosonographic evidence of necrosis | | |
| Yes | 5/5, 100% | 0.65 |
| No | 24/25, 96.0% | |
| Histologic subtype of NSCLC | | |
| Adenocarcinoma | 26/27, 96.3% | 0.74 |
| Squamous cell cancer | 3/3, 100% | |
| Number of samples sent to the pathologist | | |
| 1 | 27/28, 96.4% | 0.79 |
| 2 | 2/2, 100% | |

EBUS-TBNA: EndoBronchial UltraSound-TransBronchial Needle Aspiration, L: left, LN: lymph node, NSCLC: non-small cell lung cancer, R: right.

## 4. Discussion

A great percentage of patients with advanced NSCLC undergoes EBUS-TBNA for diagnosis and staging at mediastinal/hilar level when lymphadenopathies are detected.

Considering PD-L1 expression, concordance between LN mediastinal EBUS-TBNA samples and surgical/TBNA in primary lesion is not obvious; however, Sakakibara et al. [21] found a good correlation between EBUS samples and surgical samples in both primary and metastatic site: this feature lets Interventional Pulmonologists free to carry out EBUS-TBNA to get diagnosis, staging and prognostic information in a single procedure.

Of great interest is the role of different aspiration techniques in EBUS-TBNA in determining the diagnostic accuracy for histopathological evaluation, including molecular biology and PD-L1 amplification assessment in lung cancer diagnosis.

Only a few randomized controlled trials comparing different aspiration techniques (suction vs. no-suction) have been performed: Casal et al. [6] demonstrated a good concordance between suction vs. no-suction aspiration in terms of adequacy (88%); similar values have also been found by Lin et al. [22].

Xin He et al. retrospectively found a better acquisition of tissue-core in the slow-pull capillary aspiration [23], whereas no significant differences between suction vs. no-suction aspiration were detected by Harris et al. in a retrospective analysis [5].

To the best of our knowledge, no studies comparing the diagnostic accuracy of different aspiration techniques in EBUS-TBNA in detecting PD-L1 expression in NSCLC have been performed. In this context, it may be important to know if the EBUS-TBNA diagnostic accuracy varies according to the aspiration technique. This is the first study with the

purpose to investigate the diagnostic accuracy of slow-capillary aspiration EBUS-TBNA in the assessment of PD-L1 in NSCLC.

The results of our study of no suction aspiration EBUS-TBNA for PD-L1 showed a diagnostic accuracy of 96.7%. This result is similar to values present in literature for EBUS-TBNA. Perrotta et al. also demonstrated that EBUS-TBNA samples are suitable for PD-L1 testing, showing an accuracy in 95% of patients [1]. Similarly, a value of ca. 90% has been found by Stoy et al. [24], Heymann et al. [25], and Wang et al. [25]. In all these studies, samples were obtained through suction with EBUS-TBNA dedicated syringe and in none of the cases slow-pull capillary aspiration was used.

Our finding may induce to suppose that, not only EBUS-TBNA is an excellent technique for molecular biology evaluation, but also that slow-capillary aspiration EBUS-TBNA should be effectively used even for PD-L1 assessment in NSCLC.

Slow-pull capillary aspiration seems to show a better accuracy when applied to former/current smokers, but caution should be used in the interpretation of this sub-analysis because it doesn't represent the primary outcome of this paper.

This study has other limitations: firstly, this study is a retrospective single-centre experience: in this context a well-powered multicentric study design with a large sample size could allow to confirm our findings and to overpass the inter-observer variability.

Secondly, it should be useful to perform a randomized controlled trial, to compare slow-capillary aspiration with suction aspiration EBUS-TBNA techniques.

Thirdly, we were unable to calculate the specificity of PD-L1 testing in the present study as PD-L1 testing is not routinely performed in non-neoplastic cells and in non-NSCLC cells because its assessment does not provide any additional information and does not guide the decision-making process in clinical scenarios other than NSCLC.

## 5. Conclusions

Although the role of EBUS-TBNA for the diagnosis of hilar/mediastinal lymphadenopathies is well established, as it represents an important investigation for tissue acquisition in patients with lung cancer and for LN staging, only few studies have examined its diagnostic accuracy in PD-L1 expression assessment in NSCLC, and, to our knowledge, this is the first study that has specifically evaluated slow-pull capillary aspiration.

In this context, these data, showing that slow-pull capillary aspiration carries a diagnostic accuracy almost equal to that one reported in literature, support the use of no-suction aspiration in EBUS-TBNA for PD-L1 testing in NSCLC.

Nevertheless, a randomized controlled trial could be useful to confirm these preliminary findings.

**Author Contributions:** Conceptualization, L.Z. and F.G.; methodology, M.B.; formal analysis, F.G.; acquisition of the data, A.D.M.B., F.M., S.G., L.Z., M.S., F.B. and G.G.; writing—original draft preparation, F.G.; writing—review and editing, L.Z., F.G. and S.G. All authors have read and agreed to the published version of the manuscript.

**Funding:** This research received no external funding.

**Institutional Review Board Statement:** The study was conducted in accordance with the Declaration of Helsinki, and approved by the Institutional Review Board (Comitato Etico Regione Marche 165/2019, approval date 30 May 2019).

**Informed Consent Statement:** All subjects gave their informed consent for inclusion before they participated in the study.

**Data Availability Statement:** The data presented in this study are available on request from the corresponding author.

**Conflicts of Interest:** The authors have no conflict of interest to declare.

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
