# Peer review of "Diagnostic Accuracy of Slow-Capillary Endobronchial Ultrasound Needle Aspiration in Determining PD-L1 Expression in Non-Small Cell Lung Cancer"

_arm, doi:10.3390/arm91010001_

Round 1

Reviewer 1 Report

Good article can be publish

Author Response

Dear reviewer, 

Reviewer #1

Good article can be published

Authors' response: We would like to thank the reviewer for appreciating our manuscript.

Reviewer 2 Report

Very nice and well written paper. Few minor corrections to the text as mentioned below and in the pdf. 

The authors present an interesting study on the novel  "no suction" technique  for  EBUS TBNA in NSCLC with encouraging results.

No major issues noted in the study.    Minor issues   Please mention the sample size in the study table 1 and 2 as well as title for table 2 for standardized presentation.  Small grammatical errors as mentioned in the PDF

Author Response

Dear reviewer,

Reviewer #2

Very nice and well written paper. Few minor corrections to the text as mentioned below and in the pdf. The authors present an interesting study on the novel "no suction" technique  for  EBUS TBNA in NSCLC with encouraging results.

No major issues noted in the study.   

Authors' response: We are grateful to the reviewer for their comments.

Minor issues   Please mention the sample size in the study table 1 and 2 as well as title for table 2 for standardized presentation.  Small grammatical errors as mentioned in the PDF peer-review-24460815.v1.pdf

Authors' response: The manuscript has been amended according to your suggestions.

Reviewer 3 Report

This study provides a potential minimally invasive method to detect LN metastasis of NSCLC. But I feel confused about some descriptions or designs:

1) In the introduction, it's necessary to mention the clinical significance of PD-L1 expression in cancer diagnosis and therapy, rather than repeatedly emphasizing that this was the first time to do this study. And this part should not be placed in the discussion.

2) The 30 NSCLC patients with metastasis were diagnosed by slow-pull capillary aspiration, or other traditional tools with an additional aspiration?

3) Did the biopsies have any other NSCLC biomarker staining besides PD-L1? Many normal cells would express PD-L1 under an inflammatory circumstance. 

4) Whether the other samples that didn't have metastases or didn't apply for the slow pull aspiration also had the PD-L1 histopathology. If so, they should also be included in the statistics.

5) In fact, PD-L1 is not a tumor marker. Strictly, it is not even a reliable indicator of response to immunotherapy. Moreover, its expression is located in the membrane, cytoplasm or nucleus may represent a completely different meaning, so what is the significance of the crude count of LN PD-L1 expression in this study?

Author Response

Dear reviewer,

Reviewer #3

This study provides a potential minimally invasive method to detect LN metastasis of NSCLC. But I feel confused about some descriptions or designs.

Authors' response: Many thanks to the Reviewer for their valuable revision of the manuscript that will improve its clarity and quality.

In the introduction, it's necessary to mention the clinical significance of PD-L1 expression in cancer diagnosis and therapy, rather than repeatedly emphasizing that this was the first time to do this study. And this part should not be placed in the discussion.

Authors' response: Thanks to the Reviewer for raising this point. We have modified the manuscript accordingly. In particular, the first paragraph of the discussion have been moved to the introduction (page 2, lines 50-65).

The 30 NSCLC patients with metastasis were diagnosed by slow-pull capillary aspiration, or other traditional tools with an additional aspiration?

Authors' response: Many thanks to the Reviewer for giving us the opportunity to clarify it. All the 30 patients included in the slow-pull capillary aspiration group have been diagnosed with a “no suction” technique and no additional aspiration techniques have been used. This point has been clarified in the manuscript (page 4, line 160) adding the following sentence “Among the 174 patients, slow-pull capillary aspiration was adopted in 60 patients (34.5%), and in 30/60 patients (50.0%) NSCLC was diagnosed using only this aspiration technique”.

Did the biopsies have any other NSCLC biomarker staining besides PD-L1? Many normal cells would express PD-L1 under an inflammatory circumstance. 

Authors' response: Thanks to the Reviewer for this punctual remark. Actually, PD-L1 is not the only biomarker. In fact, considering clinical context and Hematossilin/Eosin staining, other biomarkers are used for NSCLC diagnosis: in particular, thyroid transcription factor 1 (TTF1) and p40 are always tested to distinguish between squamous cell carcinoma and adenocarcinoma; moreover, chromogranin A and synaptophysin are used in addition to hematossilin/eosin morphologic assessment to identify neuroendocrine differentiation in small cell lung cancer (SCLC) / large cell neuroendocrine carcinoma (LCNEC).  However, PD-L1 is tested to establish whether immunotherapy is feasible in NSCLC. Furthermore, PD-L1 was evaluated only in cells showing dysplastic phenotypes.

The following sentence has been added to methods to clarify this point (page 3, lines 119-128): “The final diagnosis of NSCLC was based on clinical context, hematoxilin/eosin staining, and other biomarkers. In particular, thyroid transcription factor 1 (TTF1) and p40 were tested to distinguish between squamous cell carcinoma and adenocarcinoma; chromogranin A and synaptophysin were used to identify neuroendocrine differentiation in small cell lung cancer (SCLC) / large cell neuroendocrine carcinoma (LCNEC). PD-L1 was tested to establish whether or not immunotherapy is feasible in NSCLC. PD-L1 expression at the cytoplasmic membrane was evaluated only in cancer cells.”

Whether the other samples that didn't have metastases or didn't apply for the slow pull aspiration also had the PD-L1 histopathology. If so, they should also be included in the statistics.

Authors' response: Thank you for pointing it out. We were unable to provide the requested information (i.e., the diagnostic accuracy of aspiration techniques other than slow-pull capillary aspiration and the diagnostic accuracy of slow-pull capillary aspiration in non-metastatic NSCLC). We have acknowledged it as a study limitation (page 6, lines 222-225).  Nevertheless, we would like to point out that the aim of our study was to evaluate the diagnostic accuracy of slow-pull capillary aspiration for molecular biology and PD-L1 testing in patients with hilar and/or mediastinal lymph node (LN) NSCLC metastasis. PD-L1 testing is not routinely performed in non-neoplastic cells and in non-NSCLC cells as in these clinical scenarios its assessment does not provide any additional information and does not guide the decision-making process. The latter sentence has been added to methods (page 3, lines 125-128).

In fact, PD-L1 is not a tumor marker. Strictly, it is not even a reliable indicator of response to immunotherapy. Moreover, its expression is located in the membrane, cytoplasm or nucleus may represent a completely different meaning, so what is the significance of the crude count of LN PD-L1 expression in this study?

Authors' response: Many thanks to the Reviewer for giving us the opportunity to clarify it. PD-L1 expression was evaluated at the cytoplasmic membrane only. In fact, immunotherapy with monoclonal antibodies blocking PD-1-/PD-L1 interaction has shown greater efficacy in cells with membrane PD-L1 expression. This information has been added to the methods (page 3, lines 124-125).

Round 2

Reviewer 2 Report

I appreciate the extensive revisions by the authors. I just have a minor suggestion - the technique is mentioned as no-suction in the What is the implication section whereas mentioned as minimal suction in the introduction. Please correct it. 

Author Response

Dear Reviewer,

Many thanks for their punctual suggestion. Manuscript has been modified accordingly.

Reviewer 3 Report

My concern has been well addressed. And I don't have any more questions. 

Author Response

Dear Reviewer,

Many thanks for appreciating our work.